# De-novo non-convulsive status epilepticus in adult medical inpatients without known epilepsy: Analysis of mortality related factors and literature review

Alba García-Villafranca[1], Lucía Barrera-López[2], Marta Pose-Bar[3], Elva Pardellas-Santiago[4], Jonathan G. Montoya-Valdés[2], Emilio Paez-Guillán[2], Ignacio Novo-Veleiro[2]*, Antonio Pose-Reino[2]

1 Internal Medicine Department, POVISA Hospital, Vigo, Pontevedra, Spain, 2 Internal Medicine Department, University Hospital of Santiago de Compostela, A Coruña, Spain, 3 Internal Medicine Department, University Hospital of Ourense, Ourense, Spain, 4 Neurophysiology Department, University Hospital of Santiago de Compostela, A Coruña, Spain

* ignacio.novo.veleiro@gmail.com

## Abstract

### Background

Non-convulsive status epilepticus (NCSE) often goes unnoticed and is not easily detected in patients with a decreased level of consciousness, especially in older patients. In this sense, lack of data in this population is available.

### Aims

The aim of the present study was to examine daily clinical practice and evaluate factors that may influence the prognosis of NCSE in non-epileptic medical inpatients.

### Methods

We conducted a retrospective analysis including patients admitted by any cause in an Internal Medicine ward. All patients with compatible symptoms, exclusion of other causes, clinical suspicion or diagnosis of NCSE, and compatible EEG were included. Patients with a previous diagnosis of epilepsy were excluded. We also conducted a literature review by searching the PubMed/Medline database with the terms: *Nonconvulsive Status* OR *Non-Convulsive Status*.

### Results

We included 54 patients, mortality rate reached 37% and the main factors linked to it were hypernatremia (OR = 16.2; 95% CI, 1.6–165.6; *P* = 0.019) and atrial fibrillation (OR = 6.7; 95% CI, 1.7–26; *P* = 0.006). There were no differences regarding mortality when comparing different diagnosis approach or treatment regimens. Our literature review showed that the main etiology of NCSE were neurovascular causes (17.8%), followed by antibiotic treatment

**Data Availability Statement:** We have provided the complete database in the paper and its Supporting Information files.

**Funding:** The authors received no specific funding for this work.

**Competing interests:** The authors have declared that no competing interests exist.

(17.2%) and metabolic causes (17%). Global mortality in the literature review, excluding our series, reached 20%.

## Discussion

We present the largest series of NCSE cases in medical patients, which showed that this entity is probably misdiagnosed in older patients and is linked to a high mortality.

## Conclusion

The presence of atrial fibrillation and hypernatremia in patients diagnosed with NCSE should advise physicians of a high mortality risk.

## Introduction

Non-convulsive status epilepticus (NCSE) refers to a subtype of status epilepticus characterized by the association of ictal activity with the development of clinical entities different from tonic-clonic seizures. This phenomenon combines non-convulsive or subclinical epileptic seizures (with specific abnormalities in the electroencephalographic register) and episodes of qualitative consciousness disturbances, without other major motor manifestations [1, 2].

If we refer to any kind of status epilepticus in any clinical context, NCSE accounts for approximately 20% to 30% of total status epilepticus cases, although the population incidence is difficult to determine and likely underestimated because of the absence of a unanimous definition and its clinical subtlety. The incidence and percentage of NCSE cases in medical inpatients is also difficult to stablish. Its global incidence has been estimated as ranging from 32 to 85 cases per 100,000 inhabitants and year [3–5]. Its incidence increases with age and has substantial morbidity and mortality risk, reaching an approximate general mortality rate of 22% [6, 7].

Because its clinical presentation is diverse, NCSE often goes unnoticed and is not easily detected in patients with a decreased level of consciousness and associated pathologies, specially during hospital admission by any cause [8]. The onset of NCSE usually begins with a slight change in the level of consciousness or behavior but the clinical spectrum of NCSE has a wide variety of nonspecific symptoms that range in severity, making it difficult to recognize. Three minimum criteria have been proposed for the diagnosis of NCSE: (1) decreasing consciousness level or another neurological deficit, (2) electroencephalogram (EEG) with typical changes of bioelectric status or continuous epileptic discharges and (3) clinical and electrical response to anticonvulsant drugs [9, 10]. Many factors have been considered for the underlying etiology of NCSE, including neurological disorders (brain lesions, encephalopathies, infections), systemic effects (metabolic disorders, systemic autoimmune diseases) and drugs. It is noteworthy that NCSE is diagnosed in many patients with no previous diagnosis of epilepsy [8]. Therefore, diagnosis of NCSE is a challenge in clinical practice. During hospital admission, it is usually diagnosed as delirium or other psychiatric disorders, which leads to inappropriate and late treatment, poorer response and higher mortality.

The pharmacological treatment of NCSE typically involves benzodiazepines (in monotherapy or in combination with other antiepileptic drugs), different antiepileptics and general anesthetic therapy [11]. A fast-acting algorithm has been developed to guide the use of

repeated doses of drugs, but it lacks adequate data to inform treatment selection for individual patients [1, 8, 12].

NCSE is associated with a morbidity of approximately 39%, and a mortality approaching 22% according to available data [3, 13]. However, it is difficult to differentiate the morbidity and mortality associated with the illness and those associated with treatment side effects and patient comorbidities [7].

NCSE has largely been overlooked, particularly in older people, and lack of data regarding the target population of our study (adult non-epileptic inpatients admitted by any cause) is available [5, 7, 14–16]. Therefore, the aim of the present study was to examine daily clinical practice and evaluate factors that may influence the prognosis of NCSE in medical non-epileptic inpatients admitted by any cause by analysis of a case series and compare it with the existing literature.

## Patients and methods

The clinical electronic history from all medical inpatients discharged from the hospital (University Hospital of Santiago de Compostela) during the study period (which comprised 4 years, from January 2015 to December 2018) and who were diagnosed with any form of epilepsy were reviewed. All patients with a correct diagnosis of NCSE during hospital admission in a medical ward at discharge were included, considering the diagnosis criteria for NCSE as follows: compatible symptoms (low level of consciousness or unexplained confusional state) considered by the responsible physician and the exclusion of other causes, clinical suspicion or diagnosis of NCSE, and compatible EEG. All diagnosis of NCSE were certified by a Neurologist and a Neurophysiology physician and all cases with any doubt regarding the diagnosis were excluded like those with a possible encephalopathy of any cause. All EEG registers were made with a portable registration system during at least 15 minutes. Continuous EEG monitoring is not available in general medical wards in our center, as in most of general hospitals in Spain, and it can only be used in very selected patients at intensive care units. Patients with a previous diagnosis of epilepsy were excluded and also patients from surgical units. We only considered medical inpatients with a diagnosis of NCSE made during hospital admission in medical wards, thus patients with diagnosis of NCSE after their admission in critical units were excluded, since they were not the objective of our study. In this sense, the potential number of patients at risk during the study period, considering the global data of our center and restricted to medical inpatients, reaches 20000 patients (about 5000 patients per year).

The study protocol was reviewed and approved by the Ethics Committee of Clinical Investigations of Galicia. We collected sociodemographic, epidemiological, clinical, radiological and biochemical variables in our database. The characteristics of the EEG and different treatment strategies were also reviewed through electronic records in the clinical history of each patient and were coded in our database.

Glomerular filtrate was coded following KDIGO guidelines criteria [17]. Previous brain injuries were classified into six categories: ischemic stroke, hemorrhagic stroke, demyelinating disease, venous brain disease, subarachnoid hemorrhage and tumors. All previous treatments were collected and those started the during hospital stay, with particular focus on those that could potentially trigger an NCSE episode, like antibiotics. Biochemical parameters were collected using the first analysis performed at admission and the one immediately preceding the NCSE episode. We considered hypernatremia values of plasmatic sodium over 145 mg/dL and hyponatremia values under 135 mg/dL. Hyperkalemia was coded with values of plasmatic potassium over 4.5 mg/dL and hypokalemia with values under 3.5 mg/dL. We considered as an alteration of transaminase values those patients with alanine aminotransferase (ALT) or aspartate aminotransferase (AST) over 45 UI/L.

When available, brain imaging data were collected and classified as normal, chronic findings or acute findings. EEG tracings at diagnosis were coded as pathognomonic or possibly pathognomic according to the following Salzburg criteria for NCSE with the EEG recordings and according to current guidelines and recommendations [9, 10].

The delay between the start of symptoms and the EEG performance was also coded. All NCSE treatments and their duration of use were coded in the database. Of patients who survived and were discharged, we considered as recovered those who had no relevant neurologic sequelae derived from an NCSE episode.

## Literature review search strategy and selection criteria

We conducted a literature review to identify observational studies published before May 2020 describing NCSE series or case reports. During this process, we searched the PubMed/Medline database with the following terms (related to any part of the title, abstract or manuscript): *Non-convulsive Status* OR *Non-Convulsive Status* and used PubMed filters to restrict the search to case reports, observational studies, clinical studies or clinical trials. We also retrieved additional series by surveying references included in published reports and by using the MEDLINE option "Related articles".

These search methods retrieved a total of 647 studies and after a critical read of all titles and abstracts, 320 of them were excluded because they did not match the main criteria of our review (series or case reports of adult medical patients with a complete clinical description and without previous diagnosis of epilepsy, cranioencephalic traumatism, history of recent neurosurgery, underlying hypoxic-ischemic encephalopathies or electroconvulsive therapy). We also excluded critical and surgical patients´ studies because this was not the objective of our analysis, since critical patients´ clinical profile differs from medical ones and could not be compared. After that, we analyzed the remaining 327 articles by a critical full reading, and 202 were discarded because of methodological problems, incomplete clinical information or series duplication. In cases of series with identifiable duplicate cases in other articles, all non-duplicated cases were included in the literature review. All studies describing a series of patients diagnosed through continuous EEG register in intensive care units were also discarded because this profile was not the objective of our study. Finally, 125 studies were included in the literature review.

## Statistics

Categorical variables are presented as absolute and relative frequencies. Continuous variables are presented as mean (standard deviation [SD]) or medians and interquartile ranges in cases of markedly abnormal distribution. Categorical variables were compared using the $\chi^2$ test or Fisher exact test when appropriate and continuous variables were compared through the Student's *t* test or the Mann–Whitney U test. Two-tailed $P < .05$ was considered as significant. The Spearman rank correlation (r) test was used to analyze correlations between ordinal variables. All variables associated with in-hospital mortality with $P < .10$ in univariate analysis were included to develop a multivariate logistic regression model. Hosmer-Lemeshow goodness of fit test was applied to select the best model. Odds ratio (OR) and 95% confidence intervals (CI) were reported. Statistical analyses were performed using SPSS software, version 20.0 (IBM Corp).

## Results

### Case series

A total of 54 patients were included. The mean age was 79 (SD = 13.5) years and 61% of the patients were men. Regarding previous comorbidities, 17 patients had known brain damage,

**Table 1. Baseline characteristics and comparison regarding gender.**

| Variable | Men (n = 33) | Women (n = 21) | P | Total (n = 54) |
|---|---|---|---|---|
| Age (years) | 84.5 (9.5) | 70 (14) | <0.001 | 79 (13.5) |
| Hypertension | 24 (73) | 9 (43) | 0.028 | 33 (61) |
| Diabetes | 12 (36) | 5 (24) | 0.333 | 17 (31.5) |
| Dyslipidemia | 13 (39) | 7 (33) | 0.653 | 20 (37) |
| Permanent atrial fibrillation | 15 (45.5) | 5 (24) | 0.108 | 20 (37) |
| Heart failure | 7 (21) | 6 (29) | 0.537 | 13 (24) |
| CKD | 31 (94) | 17 (81) | 0.139 | 48 (89) |
| Previous seizures | 5 (15) | 5 (24) | 0.425 | 10 (18.5) |
| Chronic brain damage | 13 (39) | 4 (19) | 0.117 | 17 (31.5) |
| Dementia | 18 (54.5) | 7 (33) | 0.128 | 25 (46) |
| Chronic psychiatric disease | 8 (24) | 5 (24) | 0.971 | 13 (24) |
| Chronic neurologic disease | 13 (39) | 9 (43) | 0.801 | 22 (41) |

Quantitative variables are shown as mean (standard deviation) and qualitative variables as absolute value (percentage). CKD: chronic kidney disease.

in most cases a previous stroke, and 10 patients had suffered a previous episode of seizures, without a formal diagnosis of epilepsy or chronic treatment to avoid recurrent episodes of seizures. Age and the presence of hypertension differed significantly between men and women. The complete profile of baseline characteristics is shown in Table 1. Considering the total number of patients potentially at risk during the study period, the incidence of NCSE in our center reached 0.27%.

With regard to the reason for the initial consultation, 22 patients (41%) referred for neurological symptoms (self-report or by their relatives), most of them for an altered level of consciousness, 12 (22%) requested clinical assistance for general status impairment and 12 (22%) complained of cardiorespiratory symptoms. Clinical examination at admission revealed a confusional state in 12 patients (22%), bradypsychia in 5 patients (9%), stupor in 5 patients and coma in 5 patients. The rest of the included patients (26; 47%) presented a normal neurological status at admission. In 33 (61%) patients, an infectious disease was the main reason for admission, in most cases these were respiratory tract infections. Other reasons for hospital admission were ischemic stroke in 6 patients (11%) and heart failure in 5 patients (9%). Only 7 patients (13%) were admitted with the diagnosis of NCSE; in the rest of the cases the diagnosis was established during the hospital stay.

Regarding concomitant drugs, 31 (57%) patients received benzodiazepines during the week prior to the NCSE episode due to sleep disorders in all cases, 19 (35%) were treated with antidepressants, 31 (57%) with neuroleptics and 52 (96%) with antibiotics, most of them beta-lactam. The reason for antibiotic treatment was respiratory tract infection in 41 cases, urinary tract infection in 9 cases and skin and soft tissue infection in 2 cases.

Examination of biochemical alterations revealed 24 (44%) patients presented with hyponatremia [mean sodium levels in this group = 132 mg/dL (SD = 8)], 8 (15%) with hypernatremia [mean sodium levels in this group = 151 mg/dL (SD = 3)], 15 (28%) with hypokalemia [mean potassium levels in this group = 3.2 mg/dL (SD = 0.7)], 15 (28%) with hyperkalemia [mean potassium levels in this group = 4.8 mg/dL (SD = 0.9)] and 28 (52%) had any elevation of liver enzymes levels.

The clinical suspicion of NCSE was established in most cases because of an unexplained decrease in level of consciousness in most patients (60%) or a persistent confusional state (19%). On the basis of these symptoms, 43 patients (80%) underwent brain computerized

tomography (CT) with acute findings in only 6 of them. EEG was performed in 44 patients (81.5%) with characteristic findings of NCSE in 30 (which completely met the Salzburg consensus criteria) and possible findings in the other 14 cases (which partially met the Salzburg consensus criteria). The diagnosis in the remaining 10 patients was stablished by clinical criteria. The decision not to perform an EEG was determined by the clinical situation of the patient, exclusion of other causes and a clinical diagnosed stablished by an expert neurologist. A lumbar puncture was performed in 8 patients with no alterations observed in any of them. Overall, there was a mean of 2.3 (SD = 4.2) days of delay between the start of compatible symptoms and the diagnosis. In 30 (55%) cases the main cause of NCSE was an ionic alteration (24 patients), frequently hyponatremia; in 19 (35%) patients the etiology of NCSE was attributed to an antibiotic treatment (all of them beta-lactams), and the other 5 (9%) patients presented an acute ischemic stroke as the main cause of NCSE. Hyperglycemia, defined as glucose levels over 200 mg/dL, were present at some point during hospital admission in 10 patients (18%), no episode of hypoglycemia was reported.

The most used treatment was levetiracetam (37 patients, 68.5%), alone or in combination, followed by benzodiazepines (27 patients, 50%), valproic acid (15 patients, 28%) and lacosamide (10 patients, 18.5%). A total of 34 patients (63%) were treated with a combined treatment and the most frequent combination was levetiracetam plus a benzodiazepine. The dosage was also very variable, in the case of levetiracetam 10 patients received 500 mg every 12 hours, 20 patients 1000 mg every 12 hours and the other 7 1500 mg every 12 hours. Valproic acid was administered by continuous perfusion in 11 cases and through intravenous bolus in the other 4 patients. We found no differences when comparing different treatment regimens regarding mortality and complete recovery rates.

The mean duration of treatment was 13 (SD = 15) days and 8 patients were transferred to an intensive care unit during hospital admission. A total of 20 (37%) patients died during the study period. The main factors linked to mortality in our series are shown in Table 2. The presence of hyperglycemia, hyper or hypokalemia or alterations in liver enzymes levels showed no differences when comparing both groups. A multivariate logistic regression analysis indicated that the main factors linked to mortality were the presence of hypernatremia (OR = 16.2; 95% CI, 1.6–165.6; $P$ = 0.019) and atrial fibrillation (OR = 6.7; 95% CI, 1.7–26; $P$ = 0.006).

A control EEG to evaluate treatment response was performed only in 7 cases (20%), in the other patients the treatment response was evaluated only by clinical criteria. Of the surviving

**Table 2. Variables linked to mortality.**

| Variable | Death (n = 20) | Survivors (n = 34) | $P$ | Total (n = 54) |
|---|---|---|---|---|
| Age (years) | 83 (11) | 77 (14) | 0.108 | 79 (13.5) |
| Men | 12 (60) | 21 (62) | 0.898 | 33 (61) |
| Permanent atrial fibrillation | 13 (65) | 7 (21) | 0.001 | 20 (37) |
| Dementia | 9 (45) | 16 (47) | 0.884 | 25 (46) |
| Chronic neurologic disease | 4 (20) | 18 (53) | 0.017 | 22 (41) |
| Chronic psychiatric disease | 1 (5) | 12 (35) | 0.012 | 13 (24) |
| Chronic neuroleptic treatment | 8 (40) | 23 (68) | 0.047 | 31 (57) |
| Hypernatremia | 7 (35) | 1 (3) | 0.001 | 8 (15) |
| Hyponatremia | 7 (35) | 17 (50) | 0.284 | 24 (44) |
| ICU admission | 1 (5) | 7 (21) | 0.119 | 8 (15) |
| Acute stroke | 1 (5) | 5 (15) | 0.395 | 6 (11) |

Quantitative variables are shown as mean (standard deviation) and qualitative variables as absolute value (percentage). ICU: intensive care unit.

patients, in most cases an antiepileptic treatment was prescribed at discharge (30 patients); the preferred drug was levetiracetam (20 patients), followed by valproic acid (9 patients) and lacosamide (4 patients). In 9 patients a combined treatment was prescribed at discharge. Follow up after discharge was performed in 17 patients for at least 1 year; none of these patients developed a new episode of NCSE during the follow up period.

## Literature review

We included 125 series and case reports (excluding ours) conducted over 35 years (1985–2020), and data from a total of 353 patients (including those from our study) were analyzed. The main data available from the related studies are shown in S1 Table. Analysis of all of the data (including our series) indicated that the etiology most commonly attributed to NCSE development were neurovascular causes (63 patients, 17.8%), followed by antibiotic treatment (61 patients, 17.2%) and metabolic causes (60 patients, 17%). The mean age of patients in the included studies (excluding ours) was 66 years old. Regarding the diagnostic process and excluding our study, 170 patients (57%) underwent brain CT, with pathological findings in 72 of them; 103 (34.5%) patients underwent magnetic resonance imaging, which showed alterations in 62 of them, and in 73 (24.5%) cases a lumbar puncture was performed, which identified pathological findings in 29 of them. Excluding our series, 60 patients (20%) died and complete recovery was achieved in 146 cases (49%). All relevant data regarding time to diagnosis, clinical management and evolution are shown in S1 Table.

## Discussion

The present study is, to our knowledge, the largest published series reporting medical non-epileptic inpatients diagnosed with NCSE during hospital admission by any cause to date. Our literature review revealed major differences from the previous series reported. Our series was composed of general inpatients admitted by any cause and reflects the reality of an internal medicine department, indicating that NCSE could be a relevant complication in these patients with a higher incidence than previously suspected. Most of the largest studies previously published included only patients admitted to critical units, with very different characteristics and which were not the target of our study [18, 19]. Because our study included only inpatients admitted for any cause, the mean age in our series was much higher than that in previous studies (79 versus 66 years old) and the presence of comorbidities was very frequent in our patients, in this sense, the acute decompensation of a chronic disease could be predisposing factor for the development of NCSE in older people [20, 21]. These differences with previous reports could have a substantial influence on the prognosis of the patients, but our approach accurately reflects the daily clinical practice of any internal medicine department, which was not reflected in the previous largest series, focused in patients who initially consulted for neurological focal symptoms, most of them with previous brain damage and admitted to Neurology departments mainly, as reflects our literature review [4, 7, 14, 22, 23].

The presence of previous brain damage was infrequent in our series, which is consistent with other series included in the literature review that have shown that brain damage does not seem to be a relevant risk factor for NCSE in older people, especially in medical inpatients [5]. The presentation of symptoms varied, but in most cases the patients showed an alteration in their level of consciousness or a persistent confusional state. Because these symptoms may be caused by multiple etiologies in older people, a high level of caution is critical in the diagnostic approach in these cases, as other authors have shown, and a proper evaluation and differential diagnosis is crucial [4, 5]. The main reason for admission in our series was infectious diseases, which is concordant with the fact that antibiotic treatment is a frequent cause of NCSE

development, reported in many cases in the literature [22, 24], according to this, antibiotic treatment was the second attributed cause for NCSE in our literature review. Despite this, our series differs from previous large reports regarding the reason for admission and reflects, in our opinion, the real daily clinical practice, as other papers have described mainly patients with a neurological disorder profile [4, 5, 7, 14, 23]. In this sense, a previous systematic review concluded that the evidence supporting the potential association of antibiotic treatment and seizure risk is low to very low [25], thus, our findings could help physicians to identify high risk patients and when to suspect a NCSE in older medical inpatients.

The diagnostic approach was performed following the current guidelines and recommendations in most cases, which reinforces the value of our findings [9]. Our literature review identified high variability in diagnostic procedures and substantial differences in the applied protocols, which makes it difficult to develop a valuable comparison that could lead to relevant conclusions (S1 Table) [7, 14, 22, 23, 26–28]. It is remarkable the variability regarding time to diagnosis, the need or not to perform complementary studies like image studies or even lumbar puncture before the diagnosis and treatment of NCSE was established. The diagnostic challenge of this entity in older patients even outweighs the technological advances and makes difficult the EEG interpretation and validation [15, 29]. In this sense, we consider that one of the main strengths of our series is the application of a diagnostic protocol mainly based on clinical suspicion an EEG criteria, which can lead to an early diagnosis and treatment [9, 10].

Ionic alterations, mainly sodium levels disturbances, were frequent in our series and were the main cause of NCSE development. This finding is concordant with previous reports included in the literature review, in which the a common etiology of NCSE were metabolic causes; in this sense, ionic alterations also have been reported as a frequent cause of NCSE [5, 30, 31] and many medical conditions and drugs can be involved in its development, which should be taken into account when diagnosing NCSE, since the correction of the primary cause could lead to the NCSE termination [5, 30–32]. These findings reinforce the importance of proper management of ionic disturbances in older people, specially hyponatremia, which frequently leads to severe complications like NCSE [33].

Beta-lactam antibiotics, specially cephalosporins, are frequently linked to the development of NCSE, although the evidence of this association remains low [25] and probably reflects a combination of risk factors present in many patients with infectious diseases. Despite this, antibiotic treatment was the second most frequent attributed cause of NCSE development in the literature review [22, 34, 35] and also in our series. These findings emphasize the importance of proper and careful use of antibiotics, mainly cephalosporins, in older people, regardless of the presence of brain damage or neurologic diseases [36].

We identified considerable variability in the specific treatments administered for NCSE, not only in the literature review, but also in our series. This variability included the type of drug, dosing, treatment duration and even the type of administration (bolus or continuous perfusion). Despite this, we found no differences in the mortality analysis related to factors of NCSE treatments. These findings suggest that the most important action with regard to NCSE management is correction of the underlying cause, especially in drug-related cases or in acute metabolic disorders, although the development of specific guidelines and treatment protocol for NCSE in older medical inpatients could have a great interest and impact in daily clinical practice.

Mortality in our series was higher than previous reports, probably because of the different profile of included patients, which reflects the reality of this severe complication in older people admitted by any cause [5, 7, 23, 37]. Additionally, the mortality in our literature review, excluding our series, was much lower than in our series. These differences can be explained through the high variability of patient baseline characteristics in previous reports, which

included patients with neurological diseases, infectious disease, dialysis treatment or substance abuse among many others [5, 7, 15, 23, 31, 37–41]. Thus, we considered that the mortality of our series is probably closer to the true mortality of NCSE in older medical patients. Moreover, our series is, to our knowledge, the first to describe variables like hypernatremia and atrial fibrillation, which could help to identify high-risk patients when diagnosing NCSE. Hypernatremia is linked to a high mortality risk, not only in patients with NCSE, but in any older inpatient [42, 43]. Atrial fibrillation is a risk factor linked to a higher mortality in older patients with any comorbidity, but its relevance in NCSE was not understood [44, 45]. In light of our findings, ionic alterations and the presence of atrial fibrillation should warn physicians of a potential high risk of mortality in patients diagnosed with NCSE. The main limitations of our case series are that the cases were limited to a single center and the analysis was retrospective in nature and also the absence of a control EEG to evaluate treatment response in most cases. Concerning the literature review, it was conducted with the aim of analyzing the variability in clinical daily practice in general medicine wards regarding NCSE and its diagnosis. The exclusion of patients diagnosed after admission in intensive care units through continuous EEG monitoring was decided to homogenize the sample and really reflect the daily clinical practice.

In conclusion, NCSE is a frequently neglected complication in inpatients admitted by any cause and has a high mortality risk. The presence of hypernatremia and atrial fibrillation could be helpful to identify high-risk patients when diagnosing NCSE.

## Supporting information

**S1 Checklist. PRISMA recommendations checklist.**
(DOCX)

**S1 Table. Cases included in the literature review by main etiology.** NP: not performed, NR: not reported, D: day, W: week, Min: minutes, LEV: levetiracetam, LCM: lacosamide, PHT: Phenytoin, MPHT: mephenytoin, VPA: Valproic acid, BZD: benzodiazepines, PB: phenobarbital, DXM: dexamethasone, DZP: diazepam, THP: thiopental, PPF: propofol, PPH: phosphophenytoin, MDZ: midazolam, LMG: lamotrigine, RPD: risperidone, HPD: Haloperidol, MTP: methylprednisolone, PDN: prednisone, CyC: cyclophosphamide, IVIG: immunoglobulins, RTX: rituximab, CLZ: clonazepam, CLP: chlorpromazine, QCN: quinacrine, PLPH: plasmapheresis, ZND: zonisamide, FBT: felbamate, PTB: pentobarbital, GBP: gabapentin, DPH: diphenylhydantoin, STE: steroids, NLX: naloxone, (ISN + RIF + PYR + ETB): isoniazid, rifampicin, pyrazinamide and ethambutol, SIRPID: stimulus-induced rhythmic, periodic, or ictal discharges, PRES: posterior reversible encephalopathy syndrome, UTI: urinary tract infection, ACS: acute coronary syndrome. *Global data from Canas N. *et al*: 6 patients admitted in ICU, 9 patients died. **Global data from Labar D. *et al*: 3 patients died and 2 presented a complete recovery.
(PDF)

**S1 File. Database.** Database including all cases with complete data.
(XLS)

## Acknowledgments

We thank Edanz Group (https://en-author-services.edanz.com/ac) for editing a draft of this manuscript.

## Author Contributions

**Conceptualization:** Alba García-Villafranca, Marta Pose-Bar, Ignacio Novo-Veleiro.

**Data curation:** Alba García-Villafranca, Lucía Barrera-López, Marta Pose-Bar, Jonathan G. Montoya-Valdés.

**Formal analysis:** Alba García-Villafranca, Lucía Barrera-López, Marta Pose-Bar, Ignacio Novo-Veleiro.

**Investigation:** Alba García-Villafranca, Lucía Barrera-López, Marta Pose-Bar, Jonathan G. Montoya-Valdés.

**Methodology:** Lucía Barrera-López, Jonathan G. Montoya-Valdés, Ignacio Novo-Veleiro.

**Project administration:** Alba García-Villafranca, Antonio Pose-Reino.

**Resources:** Lucía Barrera-López, Elva Pardellas-Santiago.

**Software:** Emilio Paez-Guillán.

**Supervision:** Emilio Paez-Guillán, Ignacio Novo-Veleiro, Antonio Pose-Reino.

**Validation:** Elva Pardellas-Santiago, Ignacio Novo-Veleiro, Antonio Pose-Reino.

**Visualization:** Antonio Pose-Reino.

**Writing – original draft:** Elva Pardellas-Santiago, Ignacio Novo-Veleiro.

**Writing – review & editing:** Antonio Pose-Reino.

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
