## [Decision Letter · Decision Letter 0]

7 Jul 2021

PONE-D-21-14243

De-novo nonconvulsive status epilepticus in adult medical inpatients without known epilepsy: analysis of mortality related factors and literature review.

PLOS ONE

Dear Dr. Novo-Veleiro,

Thank you for submitting your manuscript to PLOS ONE. After careful consideration, we feel that it has merit but does not fully meet PLOS ONE’s publication criteria as it currently stands. Therefore, we invite you to submit a revised version of the manuscript that addresses the points raised during the review process.

A rebuttal letter that responds to each point raised by the academic editor and reviewer(s). You should upload this letter as a separate file labeled 'Response to Reviewers'.A marked-up copy of your manuscript that highlights changes made to the original version. You should upload this as a separate file labeled 'Revised Manuscript with Track Changes'.An unmarked version of your revised paper without tracked changes. You should upload this as a separate file labeled 'Manuscript'

We look forward to receiving your revised manuscript.

Kind regards,

Tai-Heng Chen, M.D.

Academic Editor

PLOS ONE

Reviewers' comments:

Reviewer's Responses to Questions

**Comments to the Author**

1. Is the manuscript technically sound, and do the data support the conclusions?

Reviewer #2: No

Reviewer #3: Yes

2. Has the statistical analysis been performed appropriately and rigorously? 

Reviewer #2: No

Reviewer #3: Yes

3. Have the authors made all data underlying the findings in their manuscript fully available?

Reviewer #2: No

Reviewer #3: Yes

4. Is the manuscript presented in an intelligible fashion and written in standard English?

Reviewer #2: Yes

Reviewer #3: Yes

5. Review Comments to the Author

Reviewer #2: In this study, Garcia-villafranca et al., reviewed 54 general medical patients over a three year period with a diagnosis of NCSE, and examined important factors that determined their outcome, finding that hyponatraemia and atrial fibrillation were significantly associated with mortality secondary to NCSE.

The study highlights a condition that should be considered promptly when diagnosing fluctuant conscious state, particularly in older patients. In addition, the literature review of over 600 papers initially is commendable. However there are some major flaws with the study that inevitably minimise the findings described, that I will discuss in turn.

Major points

1. It appears the initial analysis, the main factor associated with NCSE was antibiotic use (96%), so a crucial question is the indication, e.g. systemic infection such as UTI versus brain specific e.g. encephalitis. Other factors that were associated with NCSE were statistically addressed individually. However a number of the factors would seem related e.g. atrial fibrillation and presence of stroke, neuroleptic use and chronic psychiatric disturbance, etc. and should not be analysed separately. In addition other important factors such as glucose level, functional status (MRS), are not presented. It appears more patients had hyponatraemia compared with hypernatraemia, despite the subsequent mortality analysis. Indeed comparing just 3 patients (deceased) to 1 (survived) is quite underpowered. Other factors such as potassium, hypertension or hepatic levels were not subsequently included in the multivariate analysis for mortality, is there any reason for this? Finally no actual values for biochemistry across the group are given, just the threshold values chosen for definition; this is very important when relating factors such as hypernatraemia to NCSE and its mortality.

2. A significant number of patients with NCSE were on benzodiazepine therapy (57%) prior to diagnosis (page 16) – why was this?

Minor points

1. In addition (page 15) 10 patients (1/5 of total number) had previous seizures although the authors state that patients with previous epilepsy were not included.

2. In the methods (page 13) what ‘characteristic’ EEG findings were observed to diagnose NSCE. Were all the of the Salzburg criteria satisfied for all patients?

Reviewer #3: This paper is a manuscript which can be a good contribution to the literature for nonconvulsive status epilepticus (NCSE). Here are my suggestions:

1) How was seizure response characterized? Were the authors able to collect standardized data on EEG response? If not, please discuss and add as limitation.

2) The Discussion needs a greater depth to compare your data to the literature.

3) We await the development of your treatment guidelines.

6. PLOS authors have the option to publish the peer review history of their article (what does this mean?). If published, this will include your full peer review and any attached files.

Reviewer #2: No

Reviewer #3: No

---

## [Author Response · Author response to Decision Letter 0]

25 Aug 2021

Reviewer #2: In this study, Garcia-villafranca et al., reviewed 54 general medical patients over a three year period with a diagnosis of NCSE, and examined important factors that determined their outcome, finding that hyponatraemia and atrial fibrillation were significantly associated with mortality secondary to NCSE.

The study highlights a condition that should be considered promptly when diagnosing fluctuant conscious state, particularly in older patients. In addition, the literature review of over 600 papers initially is commendable. However there are some major flaws with the study that inevitably minimise the findings described, that I will discuss in turn.

Major points

Point 1.

It appears the initial analysis, the main factor associated with NCSE was antibiotic use (96%), so a crucial question is the indication, e.g. systemic infection such as UTI versus brain specific e.g. encephalitis. Other factors that were associated with NCSE were statistically addressed individually. However a number of the factors would seem related e.g. atrial fibrillation and presence of stroke, neuroleptic use and chronic psychiatric disturbance, etc. and should not be analysed separately. In addition other important factors such as glucose level, functional status (MRS), are not presented. It appears more patients had hyponatraemia compared with hypernatraemia, despite the subsequent mortality analysis. Indeed comparing just 3 patients (deceased) to 1 (survived) is quite underpowered. Other factors such as potassium, hypertension or hepatic levels were not subsequently included in the multivariate analysis for mortality, is there any reason for this? Finally no actual values for biochemistry across the group are given, just the threshold values chosen for definition; this is very important when relating factors such as hypernatraemia to NCSE and its mortality.

Authors responses to point 1: first of all, we would like to thank the reviewer for spending her/his time in reviewing our manuscript. Regarding to the first points we have made the following changes:

-Regarding antibiotic treatment we have specified the reason for these treatment in the new version of our manuscript, as follows: The reason for antibiotic treatment was respiratory tract infection in 41 cases, urinary tract infection in 9 cases and skin and soft tissue infection in 2 cases.

-With regard to the main factors included in the mortality analysis, we referred to chronic previous conditions, also in the case of atrial fibrillation, we have changed it in Table 1 to a better comprehension.

-We have added data regarding glucose levels, unfortunately, due to the retrospective nature of the analysis we cannot add reliable data regarding to functional status.

-We agree with the reviewer that the statistical power of our analysis is poor, due to the number of cases included in each group, we have added a paragraph in the limitations section remarking this fact. Nevertheless, we still consider our results as valuable, if we take into account the number of patients included in previous reports.

-The presence of potassium levels disorders or liver enzymes alterations were not included in the multivariate analysis because the univariate analysis showed no difference between groups, we have clarified this fact in the new version of our manuscript. We have added hyponatremia in Table 1 to a better comprehension.

-Regarding actual values, we have added the mean values for patients with ionic alterations following the reviewer´s suggestion.

Point 2. 

A significant number of patients with NCSE were on benzodiazepine therapy (57%) prior to diagnosis (page 16) – why was this?

Authors responses to point 2: we have reviewed this information and the reason was the presence of sleep disorders in all cases, we have added this information in the new version of the manuscript.

Minor points

1. In addition (page 15) 10 patients (1/5 of total number) had previous seizures although the authors state that patients with previous epilepsy were not included.

Authors response: we thank the reviewer for this observation, we have clarified this fact in the manuscript. We excluded all patients with a formal diagnosis of epilepsy, this group of ten patients presented at least one isolated episode of previous seizures but had not been diagnosed with epilepsy.

2. In the methods (page 13) what ‘characteristic’ EEG findings were observed to diagnose NSCE. Were all the of the Salzburg criteria satisfied for all patients?

Authors response: we have confirmed with the neurophysiology department that all patients with a characteristic EEG satisfied the Salzburg criteria. We added this information in the new version of the manuscript.

Reviewer #3: This paper is a manuscript which can be a good contribution to the literature for nonconvulsive status epilepticus (NCSE). Here are my suggestions:

1) How was seizure response characterized? Were the authors able to collect standardized data on EEG response? If not, please discuss and add as limitation.

Authors response: first of all, we would like to thank the reviewer for spending her/his time in reviewing our manuscript. Unfortunately, a control EEG was only performed in 7 cases, we have added this data and also added a sentence regarding it in the limitations section, following the reviewer´s suggestion.

2) The Discussion needs a greater depth to compare your data to the literature.

Authors response: we thank the reviewer for this suggestion, following it we have added some information to the discussion section and made changes to improve it.

3) We await the development of your treatment guidelines.

Authors response: thank you for your words, we will be pleased to share them with you and all the scientific community in a new future paper.

---

## [Decision Letter · Decision Letter 1]

1 Oct 2021

De-novo nonconvulsive status epilepticus in adult medical inpatients without known epilepsy: analysis of mortality related factors and literature review.

PONE-D-21-14243R1

Dear Dr. Novo-Veleiro,

We’re pleased to inform you that your manuscript has been judged scientifically suitable for publication and will be formally accepted for publication once it meets all outstanding technical requirements.

Kind regards,

Emilio Russo

Academic Editor

PLOS ONE

Additional Editor Comments (optional):

Reviewers' comments:

Reviewer's Responses to Questions

**Comments to the Author**

1. If the authors have adequately addressed your comments raised in a previous round of review and you feel that this manuscript is now acceptable for publication, you may indicate that here to bypass the “Comments to the Author” section, enter your conflict of interest statement in the “Confidential to Editor” section, and submit your "Accept" recommendation.

Reviewer #2: All comments have been addressed

Reviewer #3: All comments have been addressed

2. Is the manuscript technically sound, and do the data support the conclusions?

Reviewer #2: Yes

Reviewer #3: Yes

3. Has the statistical analysis been performed appropriately and rigorously? 

Reviewer #2: Yes

Reviewer #3: Yes

4. Have the authors made all data underlying the findings in their manuscript fully available?

Reviewer #2: Yes

Reviewer #3: Yes

5. Is the manuscript presented in an intelligible fashion and written in standard English?

Reviewer #2: Yes

Reviewer #3: Yes

6. Review Comments to the Author

Reviewer #2: The authors have adequately addressed my previous comments, in particular inclusion criteria for the study and associated limitation of the relatively small cohort being analysed.

Reviewer #3: (No Response)

7. PLOS authors have the option to publish the peer review history of their article (what does this mean?). If published, this will include your full peer review and any attached files.

Reviewer #2: No

Reviewer #3: No

---

## [Editor Report · Acceptance letter]

8 Oct 2021

PONE-D-21-14243R1 

De-novo nonconvulsive status epilepticus in adult medical inpatients without known epilepsy: analysis of mortality related factors and literature review. 

Dear Dr. Novo-Veleiro:

I'm pleased to inform you that your manuscript has been deemed suitable for publication in PLOS ONE. Congratulations! Your manuscript is now with our production department. 

Kind regards, 

on behalf of

Prof Emilio Russo 

Academic Editor

PLOS ONE